# Deep Learning Using CT Images to Grade Clear Cell Renal Cell Carcinoma: Development and Validation of a Prediction Model

**DOI:** 10.3390/cancers14112574

**Published:** 2022-05-24

**Authors:** Lifeng Xu, Chun Yang, Feng Zhang, Xuan Cheng, Yi Wei, Shixiao Fan, Minghui Liu, Xiaopeng He, Jiali Deng, Tianshu Xie, Xiaomin Wang, Ming Liu, Bin Song

**Affiliations:** 1The Quzhou Affiliated Hospital of Wenzhou Medical University, Quzhou People’s Hospital, Quzhou 324000, China; qz1109@wmu.edu.cn (L.X.); fengzhang@wmu.edu.cn (F.Z.); 2Yangtze Delta Region Institute (Quzhou), University of Electronic Science and Technology of China, Quzhou 324000, China; chunyang@std.uestc.edu.cn (C.Y.); cs_xuancheng@std.uestc.edu.cn (X.C.); shixiaofan@std.uestc.edu.cn (S.F.); minghuiliu@std.uestc.edu.cn (M.L.); dengjiali@std.uestc.edu.cn (J.D.); tianshuxie@std.uestc.edu.cn (T.X.); xmwang@uestc.edu.cn (X.W.); csmliu@uestc.edu.cn (M.L.); 3University of Electronic Science and Technology of China, Chengdu 610000, China; 4West China Hospital, Sichuan University, Chengdu 610000, China; drweiyi057@scu.edu.cn; 5Affiliated Hospital of Southwest Medical University, Luzhou 646000, China

**Keywords:** clear cell renal cell carcinoma, deep learning, tumor grading, self-supervised learning, label noise, class imbalance

## Abstract

**Simple Summary:**

Clear cell renal cell carcinoma (ccRCC) pathologic grade identification is essential to both monitoring patients’ conditions and constructing individualized subsequent treatment strategies. However, biopsies are typically used to obtain the pathological grade, entailing tremendous physical and mental suffering as well as heavy economic burden, not to mention the increased risk of complications. Our study explores a new way to provide grade assessment of ccRCC on the basis of the individual’s appearance on CT images. A deep learning (DL) method that includes self-supervised learning is constructed to identify patients with high grade for ccRCC. We confirmed that our grading network can accurately differentiate between different grades of CT scans of ccRCC patients using a cohort of 706 patients from West China Hospital. The promising diagnostic performance indicates that our DL framework is an effective, non-invasive and labor-saving method for decoding CT images, offering a valuable means for ccRCC grade stratification and individualized patient treatment.

**Abstract:**

This retrospective study aimed to develop and validate deep-learning-based models for grading clear cell renal cell carcinoma (ccRCC) patients. A cohort enrolling 706 patients (*n* = 706) with pathologically verified ccRCC was used in this study. A temporal split was applied to verify our models: the first 83.9% of the cases (years 2010–2017) for development and the last 16.1% (year 2018–2019) for validation (development cohort: *n* = 592; validation cohort: *n* = 114). Here, we demonstrated a deep learning(DL) framework initialized by a self-supervised pre-training method, developed with the addition of mixed loss strategy and sample reweighting to identify patients with high grade for ccRCC. Four types of DL networks were developed separately and further combined with different weights for better prediction. The single DL model achieved up to an area under curve (AUC) of 0.864 in the validation cohort, while the ensembled model yielded the best predictive performance with an AUC of 0.882. These findings confirms that our DL approach performs either favorably or comparably in terms of grade assessment of ccRCC with biopsies whilst enjoying the non-invasive and labor-saving property.

## 1. Introduction

Renal cell carcinoma (RCC) is one of the most common deadly tumors in the urinary system, originating from the renal parenchymal urinary tubule epithelial system, accounting for 4% of human malignant tumors [1]. Clear cell renal cell carcinoma (ccRCC) is the most common subtype of RCC, accounting for about 75% of all RCC cases [2]. The Fuhrman grading system is highly recognized in the clinical oncology community, and it is widely used for diagnosing the pathological grade of ccRCC. In the Fuhrman grading system, the tumor is classified into one of four different grades (I, II, III, and IV) [3], with higher grades indicating a more serious patient condition. However, to obtain the pathological grade, the biopsy is most often carried out using a sharp tool to remove a small amount of tissue. Inevitably, this invasive procedure may entail great pain physically and mentally, whilst imposing a heavy economic burden on patients’ families and society. Recent study [4] also demonstrated that biopsy may increase the risk of complications, including hemorrhage, infection, even tumor rupture. Furthermore, considering the shortage of specialized doctors and conceivable poor conditions of equipment in some rural areas, patients in these areas may be unable to receive timely and appropriate treatment.

In recent years, deep learning (DL) has defined state-of-the-art performance in many computer vision tasks, such as image classification [5], object detection [6,7], and segmentation [7]. DL models will perform satisfactorily once they have learned enough and high-quality data [8]. Thus, given sufficient data, the accuracy of a deep-learning-enabled diagnosis system often matches or even surpasses the level of expert physicians [9,10]. A myriad of studies have validated the utility of DL in various clinical settings through various experiments, including the reduction of false-positive findings in the interpretation of breast ultrasound exams [11], the detection of intensive care unit patient mobilization activities [12], and the improvement of medical technology [13]. In the same way, DL enables the ability to non-invasively and automatically assess the pathological grade for ccRCC, monitor patients’ conditions and construct personalized subsequent treatment strategies.

However, to better apply the DL model, there are a few problematic issues that should not be lightly dismissed. First, the domain shift problem. In most deep-learning-enabled medical system, transfer learning is a common practice [14], where researchers use models pretrained on some other dataset, such as ImageNet [15]. Although ImageNet contains a large variety of images, they are all based on real-life situations and do not overlap with medical images in terms of content. The shifts between two datasets represent that the pattern-recognition abilities acquired from large datasets may not apply well to our medical task. Second is the noisy label problem [16]. Inevitably, there are always some cancerous lesions that come from high-grade patients but do not exhibit characteristics sufficient to discriminate them from low-grade patients, resulting in the mismatch between the manual labels and the actual labels. Third, the imbalance dataset problem. In most medical tasks, images for the abnormal class might be challenging to find. Developing on such an unbalanced dataset can wreak havoc on the utility of the DL model. To combat these issues, our study explores a new DL framework initialized by a self-supervised pre-training method, developed with the addition of mixed loss strategy and sample reweighting to identify patients with high grade for ccRCC.

There are also several studies related to that of ours. Zhu and collaborators [17] proposed a system that can accurately discriminate between five related classes, including clear cell RCC, papillary RCC, chromophobe RCC, renal oncocytoma, and normal, based on digitized surgical resection slides and biopsy slides. Different from this, we only focus on the ccRCC and try to explore a non-invasive tool to replace biopsy whilst providing grade assessment. Zheng [18], Cui [19], and Gao [20] had the same intention with us but their works are mainly based on radiomics, which requires using a high-throughput feature extraction method and a series of data-mining algorithms [21,22]. By contrast, our work does not need to use additional procedures, such as feature extraction, which could save labor to some extent. Most relates to our work is that of [14] which also attempted to use the deep learning model to predict the Fuhrman grade of ccRCC patients. However, it is worth nothing that this study still used ImageNet pretraining and did not pay attention to the noise and imbalance problem that may induce performance degradation in most of cases, while our framework provides a new solution to these issues with the addition of the proposed mixed loss strategy and sample reweighting, providing increased power to the common practice. To the best of our knowledge, our study is the first attempt to identify the pathological grades of patients with ccRCC in the context of a large population whilst dealing with the domain shift problem and the noisy label problem, as well as the imbalance dataset problem, simultaneously.

The specific objective of this study was to develop and validate a new DL framework to identify patients with a high grade for ccRCC based on CT images, and the results indicate that it is feasible. In addition to the application of deep learning to ccRCC pathology grading [14], we focused on the solution of these three problems. To improve the network’s capabilities, we proposed an innovative self-supervised pre-training methodology, as well as mixed loss strategy and sample reweighting to address label noise and class imbalance problems. To develop and validate our framework, we applied a temporal split to teledermatology cases: the first 83.9% of the cases (years 2010–2017) for development and the last 16.1% (years 2018–2019) for validation as done in [23]. Putting patients with different years into different groups could help avoid the bias that possibly stems from the machines and radiologic technologists, thereby being also a good practice to demonstrate the generalization ability of our method. In addition, to improve the model generalization ability, we combined several excellent single models, which achieved more reliable results. This project provides a convenient, harmless and accurate opportunity for Fuhrman grading, which will not only relieve patients from suffering from biopsies, but also assist radiologists in making diagnostic decisions in routine clinical practice, even for some rural areas.

## 2. Materials and Methods

The institution’s research ethics board approved our study. The ethics board waived informed consent because the data were obtained from preexisting institutional or public databases.

### 2.1. Patient Cohort

The patient cases covered in this study are all from West China Hospital, with a total case load of 759. We excluded 53 patients for the following reasons: (1) the CT images were incomplete or had poor image quality (*n* = 24); (2) patients with incomplete indicators (*n* = 29). Therefore, 706 patients were finally enrolled in this study. All 706 patients were admitted to the hospital from April 2010 to January 2019. From the perspective of the time domain, we assigned a total of 592 patients before year 2018 as the development cohort and a total of 114 patients after year 2018 (including 2018) as the validation cohort according to the acquisition date of the CT images. The characteristics of the included patients are shown in Table 1.

All of the pathological ccRCC patients’ grades were reconfirmed by three independent pathologists with extensive pathology experience. The labels of CT images in the validation cohort were verified by professional pathologists. This study employed the Fuhrman grading system as the benchmark. Grades I and II were assessed as low-grade, and grades III and IV were assessed as high grade. Usually, low grade has a better prognosis than high grade [24].

### 2.2. Image Acquisition

All CT scans used in this study were obtained by one of the six different CT scanners. The PCP, CMP, and NP of the MDCT (multidetector CT) examination were acquired for each ccRCC patient with strict rules. A total of 70–100 mL contrast agents were injected into the antecubital vein using a high-pressure injector at a rate of 3.5 mL/s. The PCP is the precontrast phase. The CMP means that the corticomedullary phase contrast-enhanced scan starting 30 s after injection. The NP means that the nephrographic phase contrast-enhanced scan starting 90 s after the injection. Spiral scanning and thinslice reconstruction were used for all three phases. The CT scanning parameters for the three phases were as follows: the voltage in the tube was 120 kV; the reconstruction thickness was 1 mm to 5 mm, and the matrix was 512 × 512. Only the CMP CT images were used as experimental data most of the time because the CT images are the clearest and most conducive to the analysis of the patient’s condition. The selection of only CMP CT images as experimental data somewhat reduces the times of model developing, which may impair the generalization of the model, but since our dataset includes a large enough number of cases, this operation does not have any impact.

### 2.3. Image Preprocessing

The original CT image contains interference information, of which only the tumor area is really valid for grading, so for each image, the region of interest (ROI) needs to be delineated. With 706 patients containing more than 12,000 CT images, it is clearly not desirable to have a radiologist process every image.

We utilized the DL models in target detection and segmentation to segment tumor regions in the renal CT images. In the detection and segmentation part of the tumor, we used VGG-16 [25] pre-trained on ImageNet [26] as the backbone for extracting features. A small number of images for detection and segmentation training were annotated by experienced doctors. The network was trained for 6000 epochs until its output converged. We used the trained network to detect and segment the tumors in the overall CT images, and the results were tested by an experienced radiologist, largely meeting the criteria. Figure 1 shows the tumor segmentation process. The segmented CT pictures eliminate interference from other bodily regions, allowing the content to be focused on the tumor area on the renal. The CT images involved in subsequent experiments (including pre-training process and developing process) refer to those after detection and segmentation processing. Since the size of the tumor area varies, the sizes of the CT images obtained by partitioning are different. We performed Resize or Padding operations before the data were entered into the network to make the image size uniform to the 224 × 224 × 3.

### 2.4. Self-Supervised Learning

We used a self-supervised learning (pre-training) method to equip the network with better awareness of the CT images before developing. In the pre-training and developing process, we used the RegNetY400MF, RegNetY800MF [27], SE-ResNet50 [28] and ResNet-101 [29]. Traditional pre-training models are often obtained by developing on ImageNet [15] and then using transfer learning to satisfy specific classification tasks. Such an approach suffers from the problem that there is segmentation between the pre-training and the actual classification task, with little correlation between the image contents. We used a simpler and more efficient approach to pre-train the network. The images we used in the pre-training are the same as those used in the developing, with the difference that during pre-training, we rotate the input image data clockwise in space in one of four ways (0°, 90°, 180°, 270°), and the images are labeled with the number of 90° of image rotation (0, 1, 2, 3), while during developing, CT images are labeled with the ccRCC grade of the relevant patient (0 for low-grade, 1 for high-grade). Such a pre-training method allows the network to develop feature extraction capability based on the developing images without revealing the original semantics of the developing images. We pre-trained different deep learning models using the stochastic gradient descent (SGD) algorithm and the common cross-entropy loss function. The DL models were finally trained for 60 epochs. The overall structure of the pre-training network is shown in the top half of Figure 2.

### 2.5. Mixed Loss Strategy

There are two pervasive problems in image classification (including medical image classification) tasks: one is the presence of label noise, and the other is the imbalanced data distribution. Both of these problems can be found in the data of our study.

Some malignant lesions that come from higher-grade patients do not exhibit enough characteristics to distinguish them from lower-grade patients, resulting in a mismatch between manual labeling and actual labeling. In simple terms, there are errors in the labels of CT images of some high-grade patients. To tackle the noise problem, we applied the mixed loss strategy similar to that in [30]. Suppose the labeled CT images dataset is $D={\left(x_{i},y_{i}\right)}_{i}^{N}$. During developing, the ordinary cross-entropy loss is as follows:(1)$$L_{CE}=-\frac{1}{N}\sum_{i=1}^{N}\sum_{j\in \{0,1\}}l_{ij}logp_{ij}$$
where $l_{ij}=1$ if $y_{i}=j$, and 0 otherwise. $p_{ij}$ is the network output probability that the *i*th sample belongs to category *j*. Since the true labels of some high-grade CT images were supposed to be low-grade., we add loss $L_{CE\_2}$ to alleviate the effect of noise in the developing process. Specifically, in the developing phase, under the assumption that the noise rate is α(0≤α≤1), the loss is as follows:(2)$$L_{total}=\alpha L_{CE\_1}+\left(1-\alpha \right)L_{CE\_2}$$
(3)$$L_{CE\_1}=-\frac{1}{N}\sum_{i=1}^{N}\sum_{j\in \{0,1\}}l_{ij}logp_{ij}$$
(4)$$L_{CE\_2}=-\frac{1}{N}\sum_{i=1}^{N}l_{i0}logp_{i0}$$
where $l_{i0}=1$ if $y_{i}=0$, and 0 otherwise. $p_{i0}$ is the network output probability that the *i*-th sample belongs to category 0 (low-grade). The larger the noise rate α, the higher the noise level. In the experiment, the noise rate was set at 0.4 for the best results, which is probably closest to the real noise rate of the data. Through the mixed loss strategy, we made the network learn from the modified data according to a certain probability in the developing process so as to achieve the effect of countering label noise.

### 2.6. Sample Reweighting

In terms of class imbalance, it is inevitable. For example, the proportion of mild patients in the cases of cancer detection is small, because cancer patients usually feel physical abnormalities in the middle or even late stage of the disease. The sample reweighting method is used to tackle this problem. In order to account for class imbalance when calculating cross-entropy loss, each class was weighed according to its frequency, with rare samples contributing more to the loss function [23]. Specifically, we assigned lower weights to the categories with a larger proportion of sample size. Since we have a bias toward the low-grade patient sample when dealing with the noise problem, we need to take this information into account when calculating the percentage of the number of low-grade and high-grade CT images. Suppose the weight of the low-grade patient sample is $\lambda_{0}$, and the weight of the high-grade patient sample is $\lambda_{1}$; the new weighted cross-entropy is
(5)$$L_{CE\_weight}=-\frac{1}{N}\sum_{i=1}^{N}\sum_{j\in \{0,1\}}\lambda_{j}l_{ij}logp_{ij}$$

By Equation (Equation 5), we made the network learn more from categories with smaller sample sizes. Finally, in order to comprehensively solve the problem of label noise and class imbalance, the overall optimization objective $L_{total\_weight}$ is
(6)$$L_{total\_weight}=-\alpha \frac{1}{N}\sum_{i=1}^{N}\sum_{j\in \{0,1\}}\lambda_{j}l_{ij}logp_{ij}-\left(1-\alpha \right)\frac{1}{N}\sum_{i=1}^{N}\lambda_{0}l_{i0}logp_{i0}$$

### 2.7. Developing

After pre-training, we obtained the DL model with feature extraction capability. Then, all models were developed iteratively and used to grade CT images of ccRCC patients.

It is worth noting that during the pre-training process, the classifiers of the models of the four networks are linear, i.e., one fully connected layer (with an avgpooling). During the developing process, we converted the classifier of the original network into nonlinear projection, which can perform more complex mapping and make the dimension reduction of the feature map smoother.

The weights of DL models were initialized from the networks that had been developed to classify four kinds of picture rotation angles (0°, 90°, 180°, 270°), except the projection part. The weights of the projection part are initialized in a common and efficient way [31]. To match the number of classes in our study, the output unit was modified to two (low-grade and high-grade). The developing process is shown in the bottom half of Figure 2.

After five epochs of warm up, the learning rate was set to 0.1 at the beginning and it varied as a cosine function. It is worth noting that the pre-trained backbone already has some feature extraction capability, unlike the untrained projection. Therefore, in the process of network developing, these two parts of the network should adopt different learning rates, i.e., a small learning rate for the backbone and a relatively larger learning rate for the projection. Specifically, we set the learning rate of the backbone to 0.1 times that of the projection. In addition, a weight decay rate of 0.0001 was set to inhibit overfitting, which can keep the weights of the neural network from becoming too large. Data augmentation, including random rotation and horizontal flipping, was performed on the development cohort to avoid overfitting, which can emulate the diversity of data observed in the real world. Four NVIDIA Tesla M40 graphics cards with 24 GB of memory were used in the development process. We used the SGD algorithm and cross-entropy loss defined in Equation (Equation 6) to develop the network. The DL model was finally developed with 100 epochs. Pytorch (1.0.1) and Python (3.5.7) were the main tools used in our experiments.

### 2.8. Validation and Statistics Analysis

After the developing phase, we used a validation cohort to check the generalizability of the developing effect of the model. Since each patient in the experiment contains multiple images, each image is calculated to obtain a probability vector, so for each patient there is a set of probability vectors. We statistically computed the group probability vector for each patient and finally obtained the grade judgment about the patient. When analyzing a patient’s condition, the focus is usually on the most severe part of the CT images, which is reasonable because it can accurately identify the patient’s condition. Therefore, in the statistical calculation for each patient, we used the highest probability of network output in each patient’s CT image as the judgment basis for grading. Suppose the *i*-th patient has *M* CT images, and the output of the model for each CT image is $g_{j}\left(j=1,2,\ldots ,M\right)$. The grading judgment $G_{i}$ is
(7)$$G_{i}=max\left(g_{1},g_{2},\ldots ,g_{M}\right)$$

During validation process, the accuracy (ACC), sensitivity (SEN) and specificity (SPC) were calculated to assess the capability of the DL model. In addition, we used the area under the receiver operating characteristic (ROC) curve (AUC) to show the diagnostic ability of the DL model in grading ccRCC patients.

### 2.9. Model Ensemble

Following the developing method described in Section 2.7, we developed a total of four classes of DL models with different structures in the development cohort. To improve the reliability of DL models, we combined models with different weights according to their performance in order to obtain a prediction that works best. During the experiment, we found that the single model performed close to each other. In order to increase the diversity of weights of different models in the process of model ensemble, we proposed an innovative weight calculation method. We used the model’s AUC as a reference for its ensemble weight specifically, as all four types of models have the same decile of AUC, and their ensemble weight is the value of their AUC after decile is removed. Then, for each patient, we weighted the four models’ outputs by different weights and summed them to obtain the patient’s final grading judgment. Our weight calculation method can make the models with relatively good performance occupy a larger weight in the ensemble process, increasing the difference between the weights of different models and achieving better ensemble results. Assume the weights of the four models are $\gamma_{1},\gamma_{2},\gamma_{3},\gamma_{4}$, and the *i*-th patient’s predictions are $G_{i1},G_{i2},G_{i3},G_{i4}$. The composite prediction $F_{i}$ is
(8)$$F_{i}=\frac{\sum_{k=1}^{4}\gamma_{k}G_{ik}}{\sum_{k=1}^{4}\gamma_{k}}$$

## 3. Results

We divided the CT images of 706 patients into a development cohort and validation cohort according to the acquisition date, where the development cohort contains 592 patients and the validation cohort contains 114 patients.

Four different kinds of networks (including ensemble model) were validated after developing according to our method, and the relevant metrics were calculated statistically; the validation results are shown in Table 2. The results show that our developing method exhibits satisfactory results on different networks, which illustrates the effectiveness of our method, and in contrast to the subsequent ablation experiments, it can also be seen that our method can effectively mitigate the label noise and class imbalance problems in the data. In addition, our ensemble method can effectively improve the prediction accuracy and enhance the reliability of DL model prediction results. This is like combining the opinions of multiple specialists in the patient’s diagnosis process to arrive at a more accurate and reliable judgment about the patient. We selected a model with good performance from each of the four types of models and recorded their receiver operating characteristic curves (ROC), as shown in Figure 3. We also recorded the DL model output probability of each patient in the validation cohort (0 for low-grade, 1 for high-grade), and the results are shown in Figure 4. For most high-grade patients, they have larger lesion areas and a more severe condition based on CT images, and are more likely to have a greater probability of network output. The CT images of some high-grade and low-grade patients are similar, and the probability of a corresponding network output is not significantly different. For low-grade patients, they are more likely to have a relatively smaller network output probability, and their CT images reflect a better condition. The percentage of patients who were graded as low grade or high grade by the ensemble model based on their Fuhrman grades (I, II, III, IV) is displayed in Figure 5. Figure 5 shows that the ensemble model can accurately classify patients in grades I and II as low grade and patients in grades III and IV as high grade, which is pathologically justified by treating grades I and II as low grade and grades III and IV as high grade because grades I and II have relatively more similar characteristics than grades III and IV, thus allowing the network to distinguish between low-grade and high-grade patients.

We also performed a series of ablation experiments to illustrate the effectiveness and necessity of each part of our proposed method. First, we conducted the baseline experiments, i.e., base model experiments without self-supervised pre-training, mixed loss strategy and sample reweighting, and the results are shown in Table 3. From Table 3, we can see that the overall performance of the base model is poor and biased toward the low-grade patients. The overall poor performance is mainly due to the lack of our self-supervised pre-training method. The feature extraction ability of the network is insufficient to accurately identify low-grade and high-grade patients, while the base models are biased toward low-grade patients because they do not solve the label noise and class imbalance problems.

Without using the mixed loss strategy and sample reweighting approaches, we performed experiments with self-supervised pretraining, and the results are shown in Table 4. Compared with the baseline, the self-supervised pre-training method effectively improves the performance of the models, but there is also the problem of excessive bias. Because of the lack of mixed loss strategy and sample reweighting approaches, the network will be more influenced by low-grade patients in the development process, i.e., the number of CT images of low-grade patients is larger than that of high-grade patients, which will make the network biased to low grade in the development process.

We conducted experiments with the addition of the mixed loss strategy and sample reweighting methods without the self-supervised pre-training, and the experimental results are shown in Table 5. From Table 5, we can see that the mixed loss strategy and sample reweighting can effectively solve the bias problem and improve the performance of the model, which is consistent with the fact that they can effectively solve the label noise and class imbalance problems. However, due to the lack of the self-supervised pre-training method, different networks exhibit a large gap in the integrated level relative to Table 2, which once again proves that our self-supervised pre-training method can effectively improve the network feature extraction capability, thus improving the overall network performance.

To validate the effect of different pre-training methods, we pre-trained the SE-ResNet50 model on ImageNet with other settings consistent with the experiments in Table 2. The experimental results are shown in Table 6. Compared with the ImageNet-based pre-training method, our proposed self-supervised pre-training method achieves better experimental results because the ImageNet dataset contains life-like images that have minimal association with the CT images during the developing process, and our proposed pre-training method allows the network to use the same images in the developing process as in the pre-training process and does not reveal the original semantics of the images, which makes the pre-training process and the developing process more relevant and thus allows the pre-training process to better assist the developing process.

We also conducted experiments to compare our method with different traditional machine learning methods [32] including support vector machine (SVM) [33,34,35], K-nearest neighbor (KNN), tecision tree [35], random forest [35], and gradient boosting [35]. The degree and tolerance of the SVM were 3 and 0.001. We set the number of neighbors in KNN to 5. For the decision tree, the minimum numbers of samples required to split an internal node and be at a leaf node are 2 and 1. The number of trees in random forest was set to 10. The learning rate of gradient boosting was 0.1, and the number of boosting stages to perform was 100. The experimental results are shown in Table 7. As we can see, our method clearly outperforms all the ML methods. It is worth noting that in our experiments, we did not introduce additional feature extraction methods for the ML methods, saving labor to a great extent while having reliable accuracy. The poor effect of ML methods may be due to the inability to deal with the potential noisy and imbalanced problem intrinsically existing in the data. By contrast, our framework explores a new way to deal with these issues with the help of the proposed mixed loss strategy and sample reweighting, providing increased power to the common practice.

## 4. Discussion

In this work, we proposed a radiologist-level diagnostic model based on DL approach that is capable of automatically grading ccRCC patients based on CT images. We improved the network’s capabilities using innovative self-supervised pre-training approaches. Based on the data in our research, we also proposed solutions to the label noise and class imbalance problems that exist in real world datasets, and the experimental results demonstrate the effectiveness and necessity of our work.

Our best-performing DL model has a high reliability with an accuracy of 88.2% AUC, 82.0% ACC, 85.5% SEN, and 75.0% SPEC. These results confirm that our DL method performs well or equivalent to biopsy in the grade evaluation ccRCC, with the characteristics of noninvasive and labor-saving, which can offer a valuable means for ccRCC grade stratification and individualized patient treatment.

There are four major advantages to our research. Above all, we pre-train the model with the same images (but different labels) as the developing process, in order to provide the network with a better knowledge of the images before developing. Compared with [36,37,38] using pre-trained models based on ImageNet, our method does not suffer from the problem of small correlation of image contents between the pre-training and developing process, and it allows the network to develop the same images during pre-training and developing without revealing the original semantics of the images.

Furthermore, label noise is the common problem in medical image datasets. The label noise problem degrades the label quality of medical images [39,40], which will make the medical image mismatch with its real label, and have a negative effect in the development of DL. Manually filtering all the samples undoubtedly raises labor costs, and it is inefficient when dealing with large datasets. We have taken the mixed loss strategy for the label noise, with no labor cost overhead but good results. The satisfactory experimental results verify that our method can make the DL model biased toward the correct samples in the development process. Obviously, the actual problem cannot be exactly the same for different datasets; for example, the noise rate differs in size from one dataset to another. Different real situations require different approaches, and we believe that our approach to the two challenges will aid future study in this area.

In addition, class imbalance also occurs frequently in medical image datasets. The class imbalance problem may negatively affect the performance of ML models [41] and DL models [42,43], as most classification methods assume an equal occurrence of different classes. To address this problem, we used the sample reweighting method, which yielded promising benefits. As can be seen from the experimental results, the sample reweighting method effectively prevents the DL model from favoring a certain category in the development process, that is, balance the contribution of samples with different quantity proportions to the loss function. We also expect that our approach of the topic of class imbalance will aid future study in this area.

Last but not leastFinally, DL models with different structures have different independent parameters and are developed to form different perceptions of the dataset. We combined the developed network models with various architectures and obtained more accurate prediction. The model ensemble approach can make up for the shortcomings of individual models in prediction, enhance the network generalization ability, and improve the reliability of results.

In terms of practical significance, our design can help patients in remote areas to further understand their individual conditions, assist doctors to make more accurate clinical judgments on patients’ conditions, and to a certain extent compensate for the lack of professional doctors and promote the treatment of patients. With sufficient and noise-free data and reliable developing, our method can reduce or even replace patient biopsy tests, giving patients a safer and more convenient way to be tested.

Despite the contributions of our study in grading ccRCC, it has some potential limitations. The one, although we used model ensemble to improve the generalization ability of the network. For the development of DL models, there are other more DL network architectures that can be utilized, such as VGGNet [25] and GoogleNet [44], but our experiments demonstrated the effectiveness of applying DL to the pathology grading of ccRCC patients. Next, although all cases included in our data are confirmed by professional doctors, there is still a certain human factor, so if our system is to be applied in practice, a large amount of quality data is needed to improve the model in order to make the results more reliable. The WHO/ISUP grading system has superseded the Fuhrman grading system in terms of prognosis assessment and interpretability [45]. Lastly, we take a uniform size operation (224 × 224 × 3) for tumor images of different sizes, which is necessary for network developing and validation, however, when such an operation is taken for images of small sizes, it may affect the original semantics of the images, which is one of the common problems in the image processing field.However, the intention of using cropped tumor is to exclude the interference of irrelevant information entailed by other normal region. Such normal regions do not contribute positively to the grading of ccRCC. On the contrary, the redundant information may also include a bias or shortcut that would otherwise enforce the model solving a problem differently than intended. For example, there is an observation that the network has learned to detect a metal token that radiology technicians place on the patient in the corner of the image field of view at the time they capture the image in [46].

For the clinical validation of our method, we also look forward to applying our algorithm to real world practice to protect patients from suffering of biopsies as many as possible. However, unfortunately, such a method needs special approval from corresponding authorities, which cannot be easily acquired within short notice. We will positively try this in our future work. In addition, we hope to research a better algorithm to solve the semantic loss problem caused by fixing all images to a uniform size in DL.

## 5. Conclusions

In this paper, we proposed a DL model that can effectively discriminate different grades of ccRCC patients. Based on the innovative self-supervised pre-training method, different semantics are assigned to the images so that the same images can be used in the pre-training and development tasks, which allows the network to have certain feature extraction capabilities before developing and does not make the pre-training task fragmented from the development task. In addition, we improved the accuracy of the model based on our proposed self-supervised pre-training method and alleviated the effects of label noise and class imbalance problems commonly found in the dataset and the necessity and effectiveness of the proposed method are proved by ablation experiments. With richer and cleaner samples and sufficient developing, the model may become a routine clinical tool to reduce the emotional and physical toll of biopsy on patients.

## Figures and Tables

**Figure 1 cancers-14-02574-f001:**
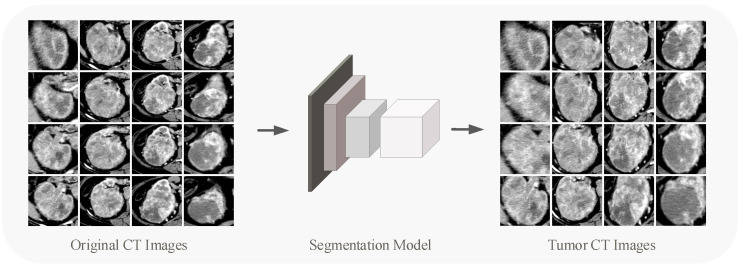
Segmentation model concentrates the CT image’s content on the tumor.

**Figure 2 cancers-14-02574-f002:**
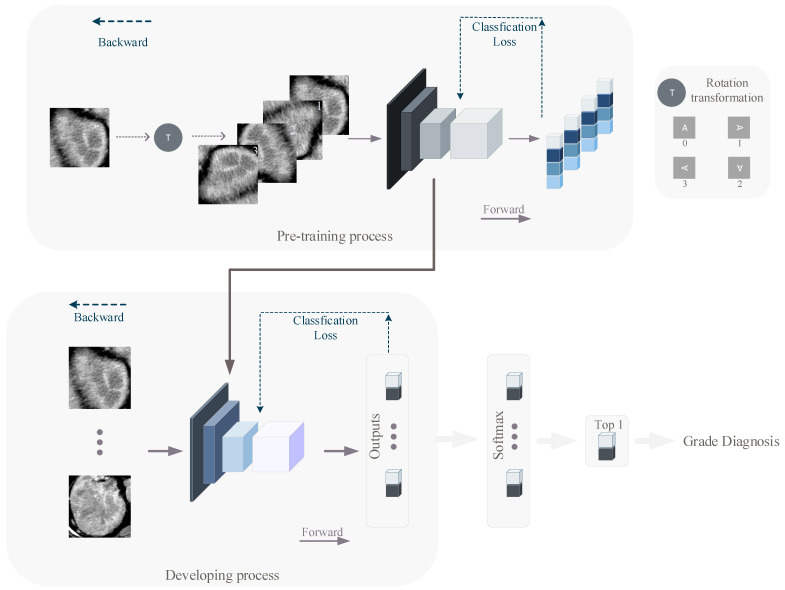
The overall flow of pre-training and developing. The top part of the figure shows the pre-training process. In the pre-training process, the original images are expanded into four images after rotation transformation, and their labels are 0, 1, 2, and 3, representing that they are obtained by quarter-turning the original image 0, 1, 2, and 3 times, clockwise. The bottom part shows the developing process. The developing process network is initialized from the pre-training process network.

**Figure 3 cancers-14-02574-f003:**
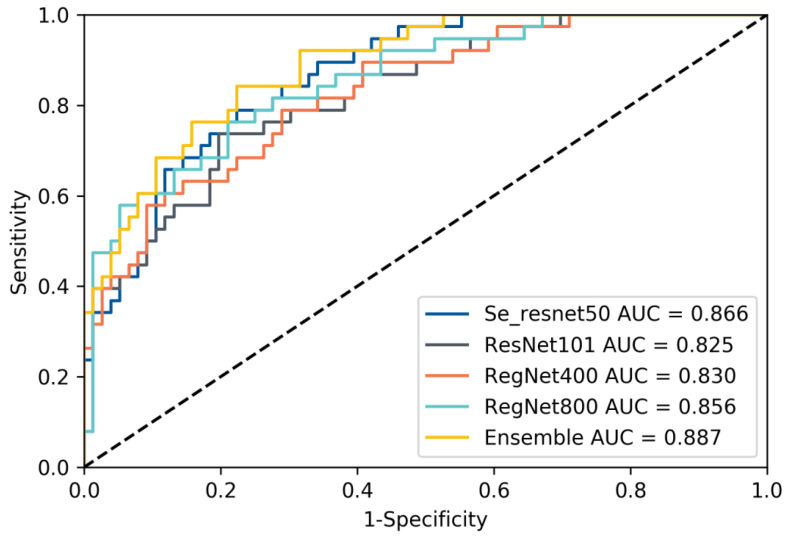
Receiver operating characteristic (ROC) curve of the four different models and the ensemble model.

**Figure 4 cancers-14-02574-f004:**
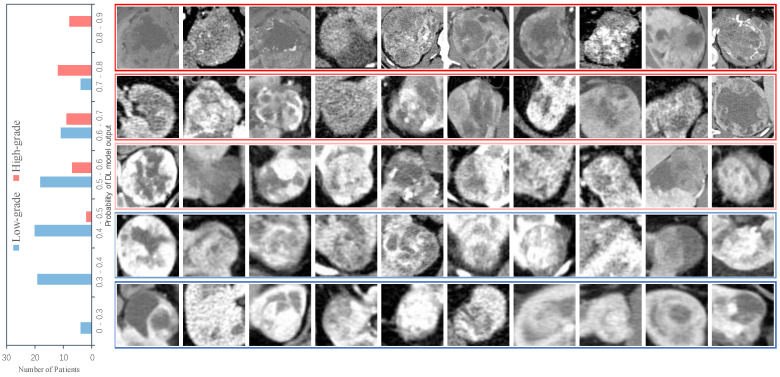
Network output probabilities for low-grade and high-grade patients. The left subplot is the network output probability distribution of low-grade and high-grade patients. The right subplot is the CT images of low-grade and high-grade patients with different network output probabilities.

**Figure 5 cancers-14-02574-f005:**
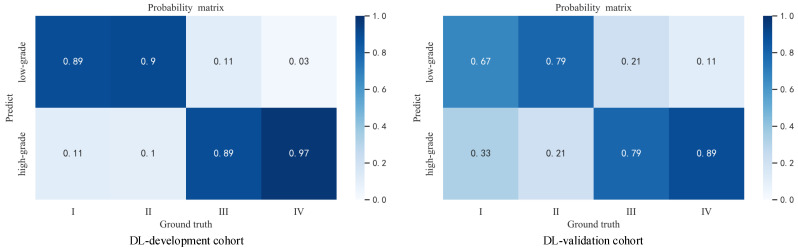
The probability matrix of four grades of patients being predicted to low-grade and high-grade. The subplot in the left is the result in the development cohort. The the subplot on the right is the result in the validation cohort.

**Table 1 cancers-14-02574-t001:** Patient characteristics.

Patient Characteristic	Development Cohort	Validation Cohort
Number	592	114
CT Images	9978	2491
Male	374 (63.2%)	71 (62.3%)
Female	218 (36.8%)	43 (37.7%)
Average Age	54.9 (±12.1)	55.8 (±12.1)
Acquisition Date	2010–2017	2018–2019
Low-grade	354 (59.8%)	76 (66.7%)
High-grade	238 (40.2%)	38 (33.3%)

**Table 2 cancers-14-02574-t002:** Results of different network models and ensemble models in the validation cohort.

Model	Sen (%)	Spec (%)	ACC (%)	AUC (%)
SE_RESNET50	85.5 ± 6.6	76.3 ± 1.3	82.5 ± 4.0	86.4 ± 0.2
RESNET101	77.6 ± 3.9	76.3 ± 4.0	77.1 ± 1.3	82.2 ± 0.3
REGNET400	82.9 ± 4.0	72.4 ± 1.3	79.4 ± 3.1	83.0 ± 0.1
REGNET800	84.2 ± 7.9	74.3 ± 4.6	81.0 ± 3.7	85.9 ± 0.3
ENSEMBLE	85.5 ± 1.3	75.0 ± 2.6	82.0 ± 0.1	88.2 ± 0.6

ACC = Accuracy; SEN = Sensitivity; SPC = Specificity; AUC = Area under the receiver operating
characteristic curve.

**Table 3 cancers-14-02574-t003:** Performance of the four basic models in the validation cohort.

Model	Sen (%)	Spec (%)	ACC (%)	AUC (%)
SE_RESNET50	65.8 ± 3.7	86.3 ± 3.4	72.6 ± 2.0	78.0 ± 2.3
RESNET101	54.4 ± 13.0	85.5 ± 12.4	64.8 ± 4.7	72.5 ± 0.6
REGNET400	65.7 ± 6.4	85.1 ± 4.5	72.2 ± 2.9	76.6 ± 0.5
REGNET800	66.4 ± 4.7	79.6 ± 2.4	70.8 ± 2.5	75.8 ± 1.4

**Table 4 cancers-14-02574-t004:** Performance of four types of self-supervised pre-trained models without mixed loss strategy and sample reweighting methods in the validation cohort.

Model	Sen (%)	Spec (%)	ACC (%)	AUC (%)
SE_RESNET50	63.1 ± 2.1	90.3 ± 2.7	72.2 ± 0.5	81.8 ± 0.8
RESNET101	68.4 ± 2.6	80.3 ± 1.3	73.4 ± 0.3	81.2 ± 0.6
REGNET400	69.3 ± 2.5	79.8 ± 1.6	72.8 ± 1.3	80.8 ± 0.2
REGNET800	62.3 ± 2.5	93.0 ± 2.5	72.5 ± 0.8	82.7 ± 0.2

**Table 5 cancers-14-02574-t005:** Performance of four types of basic models with mixed loss and sample reweighting methods in the validation cohort.

Model	Sen (%)	Spec (%)	ACC (%)	AUC (%)
SE_RESNET50	76.2 ± 3.6	75.0 ± 1.1	75.9 ± 2.2	79.2 ± 1.1
RESNET101	73.7 ± 2.1	76.8 ± 3.3	74.7 ± 1.1	80.4 ± 0.3
REGNET400	72.8 ± 8.9	73.2 ± 10.6	72.9 ± 2.5	79.4 ± 1.1
REGNET800	75.0 ± 2.3	75.3 ± 3.0	75.1 ± 0.6	80.0 ± 0.7

**Table 6 cancers-14-02574-t006:** Comparison of the SE-ResNet50 model performance based on different pre-training methods in the validation cohort.

Model	Sen (%)	Spec (%)	ACC (%)	AUC (%)
ImageNet	75.0 ± 1.3	77.3 ± 3.3	75.7 ± 1.2	80.3 ± 0.8
Ours	85.5 ± 6.6	76.3 ± 1.3	82.5 ± 4.0	86.4 ± 0.2

**Table 7 cancers-14-02574-t007:** Performance of machine learning methods in the validation cohort.

Model	Sen (%)	Spec (%)	ACC (%)	AUC (%)
SVM	63.2 ± 18.9	63.2 ± 17.8	63.2 ± 6.7	62.5 ± 7.1
KNN	71.2 ± 16.0	54.6 ± 17.9	60.3 ± 7.1	65.2 ± 2.9
DecisionTree	96.1 ± 2.9	12.8 ± 3.4	40.1 ± 1.6	54.4 ± 1.0
RandomForest	61.8 ± 7.8	68.8 ± 5.7	66.4 ± 1.9	68.4 ± 3.1
GradientBoosting	63.8 ± 11.7	75.7 ± 13.0	71.7 ± 5.9	68.7 ± 4.1
Ours-Ensemble	85.5 ± 1.3	75.0 ± 2.6	82.0 ± 0.1	88.2 ± 0.6

## Data Availability

The de-identified CT images data used in this study are not publicly available due to restrictions in the data-sharing agreement.

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
