# Peer review of "Deep Learning Using CT Images to Grade Clear Cell Renal Cell Carcinoma: Development and Validation of a Prediction Model"

_cancers, 2022, doi:10.3390/cancers14112574_

Round 1

Reviewer 1 Report

The authors proposed a deep learning-based self-supervised pre-training methodology
approach to grade clear cell renal cell carcinoma (ccRCC) from CT images. However, the
author needs to focus on some of the points identified as follows.
1. Novelty:
The novelty is in question. There are many similar methods for ccRCC classification.
Some of these are shown below:
1.1 Zhu, Mengdan, et al. "Development and evaluation of a deep neural network for
histologic classification of renal cell carcinoma on biopsy and surgical resection
slides." Scientific reports 11.1 (2021): 1-9.
1.2 Zheng, Zaosong, et al. "Development and validation of a CT-based nomogram for
preoperative prediction of clear cell renal cell carcinoma grades." European
radiology 31.8 (2021): 6078-6086.
1.3 Cui, Enming, et al. "Predicting the ISUP grade of clear cell renal cell carcinoma
with multiparametric MR and multiphase CT radiomics." European
Radiology 30.5 (2020): 2912-2921.
1.4 Gao, Rui-Zhi, et al. "Development and Validation of a Radiomics Nomogram
Model for Predicting the Prognosis of Kidney Renal Clear Cell
Carcinoma." Frontiers in oncology 11 (2021): 2347.
A comprehensive study of the previous latest papers should be there, which clearly
points out their shortfalls and research gaps thereby analysing the novelty of the topic
and methodologies by adding a literature review section. Then, the motivation for
proposing the model is described more convincingly and reasonably.
2. Cross-validation:
The author does not consider the cross-validation method to validate their results. K-5
or K-10 cross-validation is the standard cross-validation protocol. Its usage and
comparison with the current results are required.
3. Data Augmentation:
The authors should clearly discuss the kind of data augmentation used in the
experiments, to make them fully reproducible.
4. Comparison with other similar work: The benchmarking table representing the
comparisons with other similar work is missing.
5. Clinical validation:
There is no clinical validation as proof of the correctness of the proposed system.

6. Implementation Requirement: The hardware implementation requirement is
provided in section 2.7, however, there is no information about the software
requirement.
7. The proposed method is validated with images having a small size that might affect
the original semantics of the images. Please provide reasonable arguments on this.
8. The authors compared several methods and listed the results. Could the authors
provide the key parameters of these methods? It is known that the set of parameters
may have huge effects on the model performance.
9. Future work and extension:
The future scope and extension of the current work should be clearly mentioned.

Reviewer 2 Report

'Clear cell renal cell carcinoma(ccRCC) pathologic grade' is very important clinical information.
Identification of pathologic grade, in advance without biopsy, will be of great help in patient treatment.

The objective and methods of this study are well defined, easy to read and well written, and the main result is also a sufficient basis for their conclusion.
It follows the standard form of AI research.

Minor revision.

Comment 1. Put a space in front of abbreviations of words.

Comment 2. Is there no previous research to predict the pathologic grading system using medical imaging? (including other medical imaging modality)
If so, add it to the first paragraph of the Introduction.
If not, emphasize that this is the first study.

Comment 3. Table 1.
Indicate % for categorical variables.
Add standard deviation value to Age.

Comment 4. The weakness of this study is that they treat all the image independently.
And then, they picked the highest score among all the slide images from a patient.
In this regard, there is a disadvantage that a high-grade loss can be given even for slide that does not reflect high-grade at all; To overcome this issue, some studies treat the input image (CT) as 3D shape to train the deep learning model, or pick a representative image by expert radiologist.

This limitation must be dealt with.

Reviewer 3 Report

The manuscript entitled “Deep Learning using CT images to grade clear cell renal cell carcinoma: development and validation of a prediction model” aims to develop and validate a novel deep learning framework able to identify patients with high grade ccRCC on Ct images. The manuscript is interesting and embraces a current interesting topic, however different issues have to be addressed before proceeding to the publication. In addition, typos and English grammar has to be thoroughly revised along the manuscript.

The following points have to be checked

  • Major issues

INTRODUCTION

Regarding the role of radiomics, improve the introduction as several studies are currently ongoing and recently published on this issue. See DOI: 10.1016/j.crad.2021.03.001 and DOI: 10.3390/ijms22189971

MATERIALS AND METHODS

Concerning the different CT scans used and the different pathologists involved, it has to be state, in the limitation of the study, that this could represent a potential bias.

Please report how did you calculate region of interest and extracted features

RESULTS

Albeit I appreciated the synthesis, results seems too rushed.

DISCUSSION

Too schematic the “first-second-third” writing. Critically analyze your findings and, most importantly, report the potential limitations of your study which are completely lacking. Moreover, future perspectives and influence on RCC treatment should be widely reported in order to provide a solid background for the conclusion of your work.

CONCLUSION

Too rushed. Check previous comments to improve the quality of your work.

  • Minor issues

INTRODUCTION

Revise introduction, improving the epidemiological data reported.

MATERIALS AND METHODS

Avoid colloquial words or terms. For example “the original CT images contain too much unimportant information”. The same issue is repeated in the subparagraph 2.5 and 2.6

Round 2

Reviewer 3 Report

in my opinion the authors answered all comments and suggestions. So the article is now suitable for publication in your journal